# Immobilized KDN Lipase on Macroporous Resin for Isopropyl Myristate Synthesis

**Ming Song, Yuhan Xin, Sulan Cai, Weizhuo Xu \*** 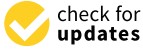 **and Wei Xu \***

School of Functional Food and Wine, Shenyang Pharmaceutical University, 103 Wenhua Road, Shenhe District, Shenyang 110016, China
* Correspondence: weizhuo.xu@syphu.edu.cn (W.X.); shxuwei8720@163.com (W.X.);
  Tel./Fax: +86-024-43520301 (W.X.); +86-024-43520307 (W.X.)

**Abstract:** Free enzymes often face economic problems because of their non-repeatability and variability, which limit their application in industrial production. In this study, KDN lipase was immobilized with the macroporous resin LXTE-1000 and glutaraldehyde. The optimal conditions of enzyme immobilization were defined by a single factor experiment and response surface methodology (RSM). The concentration of the cross-linking agent glutaraldehyde was 0.46% (*v/v*), the cross-linking temperature was 25.0 °C, and the cross-linking time was 157 min. The enzyme activity of the immobilized KDN lipase after adsorption/cross-linking was 291.36 U/g, and the recovery of the enzyme activity was 9.90%. The optimal conditions for the synthesis of isopropyl myristate were catalyzed by the immobilized KDN lipase in a solvent-free system: immobilized enzyme 53 mg, reaction temperature 36.1 °C, myristic acid 228.4 mg, isopropanol 114 μL, and reaction time 18 h. The yield of isopropyl myristate was 66.62%. After ten cycles, the activity of the immobilized KDN lipase preserved more than 46.87% of its initial enzyme activity, and it demonstrated high tolerance to solvents compared to free KDN lipase.

**Keywords:** lipase; adsorption; cross-linking; immobilization; response surface methodology (RSM)



## 1. Introduction

As a useful substitute and supplement to chemical catalysis, biocatalysis has been widely used in various fields. It has developed into a mature technology in chemical and pharmaceutical synthesis and other fields requiring mild reaction conditions and high selectivity [1]. Lipase can hydrolyze triacylglycerol (TAG) into glycerol and free fatty acids and catalyze many chemical reactions, such as esterification and transesterification [2]. In a non-aqueous medium, lipase can synthesize glycerol and long-chain fatty acids into esters [3]. Moreover, lipase is characterized by a high yield, wide temperature range, stability in the pH range, and the absence of cofactors [4]. Although lipase has many advantages, free enzymes have a low tolerance to organic solvents, they are expensive, and they have no economic prospects. Unlike free enzymes, immobilized enzymes are insoluble in reaction mixtures. Therefore, they can be easily removed and recycled from the system [5,6]. At the same time, a good immobilization strategy has improved the stability of lipase in harsh reaction environments [7]. In recent years, enzyme immobilization methods have developed rapidly, mainly including adsorption, cross-linking, encapsulation, and covalent binding [8,9]. The immobilization of lipases on solid carriers has been a hot topic in the field of immobilized enzymes to overcome the limitations of the homogeneous catalytic process. If the immobilization method is determined, the selection of suitable carrier materials is another key aspect to achieve the optimal enzyme immobilization [10]. Generally, the carrier structure needs to be resistant to chemical attacks, deformation, and microbial decomposition. Additionally, the surface of the carrier must be rich in active groups and hydrophilic chains [11]. In the research, LXTE-1000 is one of the newly developed resins with strong mechanical strength, a suitable pore size, and a large surface area.

The discovery of non-aqueous enzymatic reactions benefited from Zaks and Klibanov's research on the catalytic behavior and thermal stability of lipase in an organic medium, which completely broke the limitation of the use of an aqueous enzymatic reaction [12]. Compared with an aqueous enzymatic reaction system, a non-aqueous enzymatic reaction system has the following advantages: Firstly, it can improve the solubility of a substrate. Most substrates are slightly soluble or insoluble in water. Secondly, it can inhibit some side reactions that generate water so that the thermodynamic equilibrium of the reaction moves forward. Finally, lipase is insoluble in organic solvents, which is convenient for the recovery and purification of the product [13].

As an important food and pharmaceutical component, isopropyl myristate exhibits the typical aliphatic ester structure, as shown in Figure 1. It is soluble in ethanol, ether, chloroform, and other organic solvents, with a melting point of 3 °C; it is miscible with vegetable oil, insoluble in water, and not easy to hydrolyze or cause rancidity [14].

**Figure 1.** Chemical structure of isopropyl myristate.

In the food industry, isopropyl myristate can be directly added to improve the aroma and taste of food. It can also be used to assist the addition and dispersion of other food additives by virtue of its low viscosity, good mutual solubility, and other advantages [15]. It is often used as a substitute for natural oil in the cosmetics industry [16]. In the pharmaceutical industry, isopropyl myristate is often used as a lubricant and thickener, and is the carrier of many active pharmaceutical ingredients [15]. In transdermal preparations, it is also used as an auxiliary solvent to enhance the penetration of active ingredients through the skin [17]. The traditional method of synthesizing isopropyl myristate usually uses chemical catalysts under high temperatures, which leads to bad changes in terms of the color, smell, and stability of the final product. Therefore, subsequent refining and purification are required. Although the yield of chemical synthesis method is high, in the whole synthesis process, strong acids and corrosive catalysts are used, and the reaction temperature is high, the energy consumption is serious, and it is not friendly to the environment [15]. Despite previous attempts to use lipase to synthesis isopropyl myristate [16,18–20], the lipase used is derived from *Bacillus*, *Aspergillus*. In this current work, a new lipase, the KDN lipase, was used to catalyze the production of isopropyl myristate. The enzymes were immobilized on a new resin, LX TE-1000, and the reactions were performed in a solvent-free system. The reactions were optimized by response surface methodology (RSM). The research results can be used as a reference for the industrial enzymatic synthesis of isopropyl myristate and have certain potential application value.

## 2. Results and Discussion

### 2.1. Optimization of Macroporous Resin Adsorption/Cross-Linking

The theoretical optimal immobilization conditions for the immobilization of KDN lipase by the adsorption method are as follows: The enzyme added is 60.00 mg/g, the adsorption temperature is 33.90 °C, the buffer's pH is 6.80, and the adsorption time is 80 min. The enzyme activity was determined to be 210.58 U/g, and the recovery rate of the enzyme activity was 9.62% (Tables S1 and S2, Figure S1). The activity of the immobilized KDN lipase was only 34.30% of its initial activity after it was used for one cycle. After seven cycles, the enzyme activity was only 4.81% of its initial enzyme activity, and the use stability was poor (Figure S2). The reason for this might be that the force between LXTE-1000 macroporous resin and KDN lipase was weak, which caused most of the KDN

lipase to fall off of the carrier. Therefore, the cross-linking agent was subsequently used for improvement.

Three cross-linking agents, namely glutaraldehyde, poly (ethylene glycol) diglycidyl ether, and ethylene glycol diglycidyl ether, were investigated. Each cross-linking agent was set up with six groups: 0.10%, 0.30%, 0.50%, 0.70%, 0.90%, and 1.00% ($v/v$), among which the control group was KDN lipase, which was only adsorbed (Figure 2A). The results show that the cross-linking effect of glutaraldehyde was the best among the three cross-linking agents and was labeled as KDN@LXTE-1000/Glutaraldehyde. The possible reason for this is that when the concentration of the cross-linking agent is too small, the cross-linking is not sufficient; when the concentration is too high, the cross-linking agent has a toxic effect on the lipase, leading to a decline in the enzyme activity. When the cross-linking temperature was 30.0 °C, the activity of the immobilized KDN lipase was the highest (Figure 2B). When the cross-linking time is between 2 and 4 h, the enzyme activity of the immobilized KDN lipase increases with the extension of the cross-linking time; during 4–8 h, the enzyme activity decreases. Therefore, the optimum cross-linking time is 4 h (Figure 2C).

The glutaraldehyde, which is the most popular cross-linking agent, can interact with different functional groups of enzymes (amine, thiol, phenol, and imidazole), but it mainly reacts with the primary amine group of lysine (Lys) residues on the enzyme surface due to its higher nucleophilicity [17,21]. This cross-linking characteristic improves the stability of immobilized enzymes. Covalent binding and cross-linking produce strong interactions, but they can alter the shape of enzyme structures, resulting in enzyme inactivation [22,23].

### 2.2. Optimization of KDN Lipase Immobilization Using RSM for Immobilization Efficiency

The second-order polynomial model was chosen to study the interaction of two different parameters on the same response [24]. The immobilization efficiency and the process variables of concentrations of the cross-linking agent ($v/v$ %), cross-linking temperature (°C), and cross-linking time (h) are linked by a second-order polynomial regression equation. A second-order polynomial model is selected to study the interaction of two different parameters on the same response. The process variables of the immobilization efficiency and cross-linking agent concentration ($v/v$ %), cross-linking temperature (°C), and cross-linking time (h) are linked. Table 1 shows that $p = 0.0135 < 0.05$, indicating that the model is significant. The model also shows a statistically insignificant lack of fit ($p = 0.3032$), indicating that the response was sufficient for the model and that it fit well with the experimental data. Therefore, this model can better explain the influence of the above three factors on the activity of the immobilized KDN lipase by the adsorption/cross-linking method. The KDN lipase immobilization efficiency and the relationship between the factors are shown in the following equation:

$$Y = 265.53 + 25.44A - 4.47B + 1.03C - 16.27AB - 9.10AC + 9.55BC - 29.48A^2 - 5.33B^2 - 11.50C^2$$

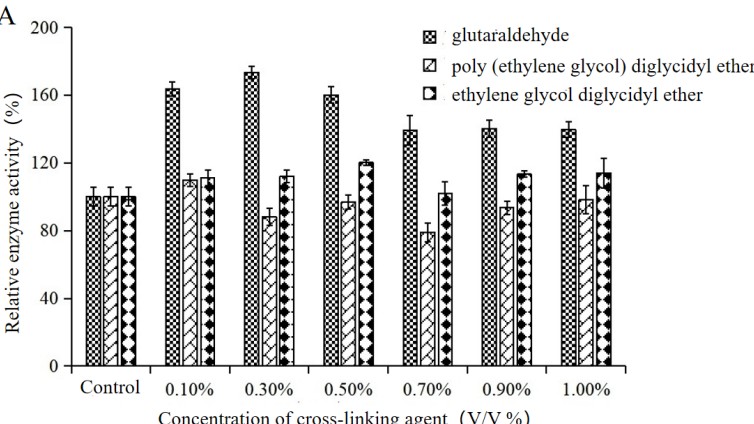

**Figure 2.** *Cont.*

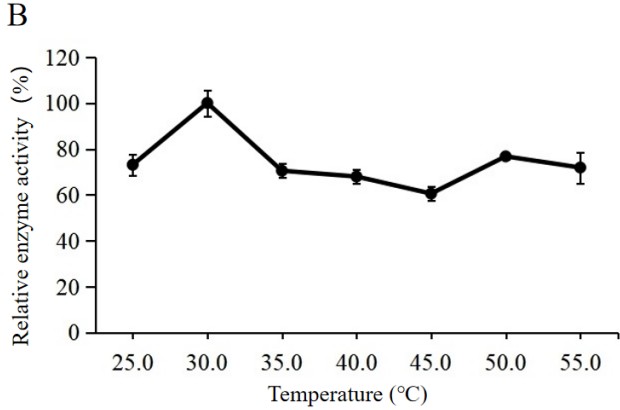

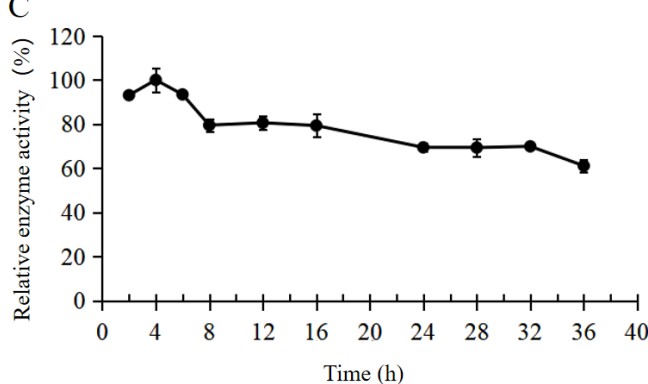

**Figure 2.** Effect of type and concentration of cross-linking agent, cross-linking temperature, and cross-linking time on immobilized lipase-specific activities. (**A**) Glutaraldehyde, poly (ethylene glycol) diglycidyl ether, and ethylene glycol diglycidyl ether were investigated. The control group was only treated with adsorption. (**B**) Cross-linking agent is glutaraldehyde. (**C**) Cross-linking agent is glutaraldehyde and enzymatic reaction temperature is 30 °C. The means value of three replicates $\pm$ SD is represented.

**Table 1.** ANOVA and fit statistics of the quadratic model for immobilization efficiency.

| Model | Sum of Squares | Mean Square | *F*-Value | *p*-Value | Significant |
|---|---|---|---|---|---|
| | 11,724.46 | 1302.72 | 6.05 | 0.0135 | |
| A | 5179.08 | 5179.08 | 24.04 | 0.0017 | very significant |
| B | 160.03 | 160.03 | 0.75 | 0.4173 | |
| C | 8.47 | 8.47 | 0.039 | 0.8485 | |
| AB | 1058.53 | 1058.53 | 4.91 | 0.0622 | |
| AC | 330.88 | 330.88 | 1.54 | 0.2552 | |
| BC | 365.00 | 365.00 | 1.69 | 0.2342 | |
| $A^2$ | 3658.37 | 3658.37 | 16.98 | 0.0045 | |
| $B^2$ | 119.80 | 119.80 | 0.56 | 0.4801 | |
| $C^2$ | 556.50 | 556.50 | 2.58 | 0.1520 | |
| Residual | 1508.07 | 215.44 | | | |
| Lack of Fit | 664.26 | 221.42 | 1.05 | 0.3032 | non-significant |
| Pure Error | 843.81 | 210.95 | | | |
| Cor Total | 13,232.53 | | | | |

Table 1 shows that the order of influence of the three factors on the immobilization of KDN lipase is as follows: the concentration of the cross-linking agent (A) > the cross-linking temperature (B) > the cross-linking time (C). Among them, the cross-linking agent has a very significant influence on the immobilization of KDN lipase ($p < 0.01$). The Design Expert

V8.0.6 software was used to analyze the interactions between various factors. The analysis results are shown in Figure 3. Combined with Table 1, it can be seen that the interaction between the concentration of cross-linking agent and the cross-linking temperature had a greater impact on the immobilization of KDN lipase by adsorption/cross-linking.

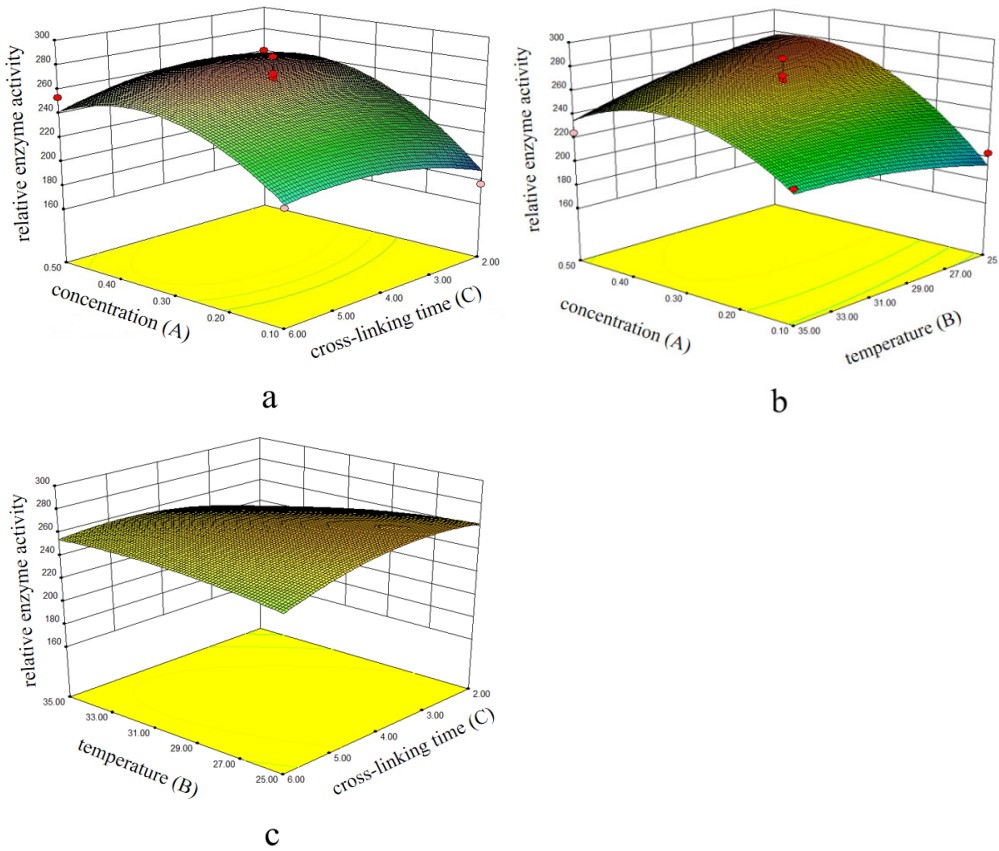

**Figure 3.** Response surface plot and immobilization efficiency. (**a**) Concentration of cross-linking agent vs. cross-linking time. (**b**) Concentration of cross-linking agent vs. cross-linking temperature. (**c**) Cross-linking temperature vs. cross-linking time.

The extreme point was obtained by derivation of the regression equation, and the theoretical optimal immobilization conditions were as follows: the concentration of glutaraldehyde was 0.46% (*v/v*), the cross-linking temperature was 25.00 °C, and the cross-linking time was 157 min. The enzyme activity of the immobilized KDN lipase by adsorption/cross-linking was determined to be 291.36 U/g, and the recovery of the enzyme activity was 9.90%.

### 2.3. Biochemical Characterization of KDN@LXTE-1000/Glutaraldehyde and Free KDN Lipase

2.3.1. Thermal and pH Stability

In industrial production, the temperature stability of an immobilized enzyme is very important. The optimum reaction temperature for the immobilization of KDN lipase by adsorption/cross-linking (KDN@LXTE-1000/Glutaraldehyde) and the immobilization of KDN lipase by adsorption (KDN@LXTE-1000) and free enzyme was 40.0 °C. When the temperature was between 40.0 °C and 80.0 °C, the enzyme activity decreased with the increase of temperature. At 70.0 °C, the residual activities for KDN@LXTE-1000/Glutaraldehyde, KDN@LXTE-1000, and the free enzymes declined to 79.00%, 61.35%, and 13.13%, respectively (Figure 4A). With the prolongation of the incubation time, the enzyme activities of KDN@LXTE-1000/Glutaraldehyde, KDN@LXTE-1000, and the free enzymes decreased. After incubation for 7 h, the free enzyme activity only retained 31.37%, KDN@LXTE-1000 re-

tained 37.44%, and KDN@LXTE-1000/Glutaraldehyde retained 76.30% of the initial enzyme activity (Figure 4B). These results showed that higher operating temperatures changed the three-dimensional configuration of the enzyme, and then affected the recovery of the free enzyme activity. The temperature tolerance of the immobilized lipase was significantly enhanced, and the adsorption material could stabilize the three-dimensional conformation of the enzyme. The loss of enzyme activity of free enzymes at high temperatures is caused by the denaturation of the enzyme. The improvements in heat resistance of the immobilized enzyme are due to its ability to prevent conformational changes caused by heat [25].

Furthermore, the pH stability for KDN@LXTE-1000/Glutaraldehyde, KDN@LXTE-1000, and free KDN lipase was determined by incubation at different pH ranges from 4 to 10. The optimum reaction pH of KDN@LXTE-1000/Glutaraldehyde, KDN@LXTE-1000, and the free enzyme was 6.0. At pH 4.0, the residual activities for KDN@LXTE-1000/Glutaraldehyde, KDN@LXTE-1000, and the free enzyme declined to 84.80%, 73.43%, and 52.25%, respectively. At pH 10.0, the residual activities for KDN@LXTE-1000/Glutaraldehyde, KDN@LXTE-1000, and the free enzyme declined to 76.15%, 67.90%, and 49.95%, respectively (Figure 4C,D). The pH has a great influence on the stability of the enzyme. It can be seen that the immobilized enzyme has the best pH stability after cross-linking. However, a continuous increase in pH can lead to a decrease in the immobilization efficiency. This may be due to changes in the enzyme's protein structure under these pH conditions [26].

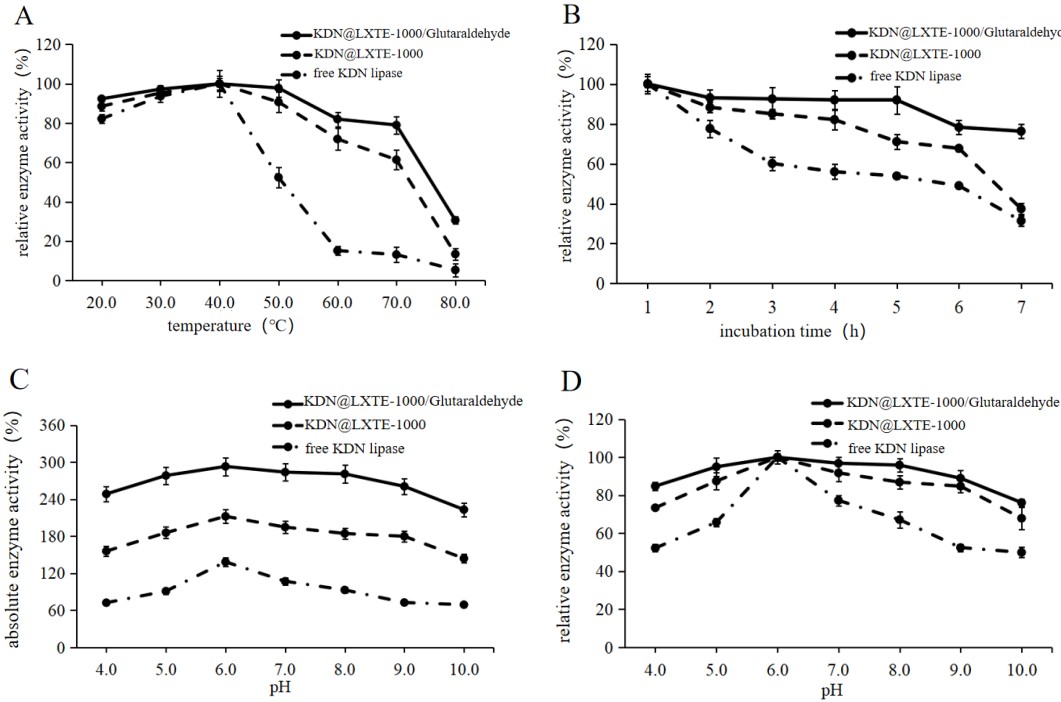

**Figure 4.** Thermal and pH stability of KDN@LXTE-1000/Glutaraldehyde, KDN@LXTE-1000, and free enzyme. (**A**) Determination of optimum temperature. (**B**) Determination of temperature stability. (**C**) Determination of pH stability (expressed with absolute enzyme activity). (**D**) Determination of pH stability (expressed with relative enzyme activity). The error bars show standard deviations; all studies were conducted in triplicate.

### 2.3.2. Solvent Tolerance, Reusability, and Storage Time Analysis

The stability of lipase in the presence of a solvent is very important. It can be seen from Figure 5A that after cross-linking, the tolerance of the immobilized KDN lipase to organic solvents, such as n-hexane, cyclohexane, isooctane, petroleum ether, isopropanol, acetone, and ethanol, is improved. The tolerance to petroleum ether only decreased. Moreover, cyclohexane and isooctane can activate the activity of the enzyme immobilized by adsorption/cross-linking (KDN@LXTE-1000/Glutaraldehyde). It can be seen that the

tolerance to organic solvents of the immobilized KDN lipase by adsorption/cross-linking is better than that of the immobilized KDN lipase by adsorption (KDN@LXTE-1000). In industrial large-scale productions, the use of organic solvents is inevitable. Therefore, it is very important that the immobilized enzyme can improve the tolerance to organic solvents, which can enable laboratory preparation to be transformed into industrial manufacturing.

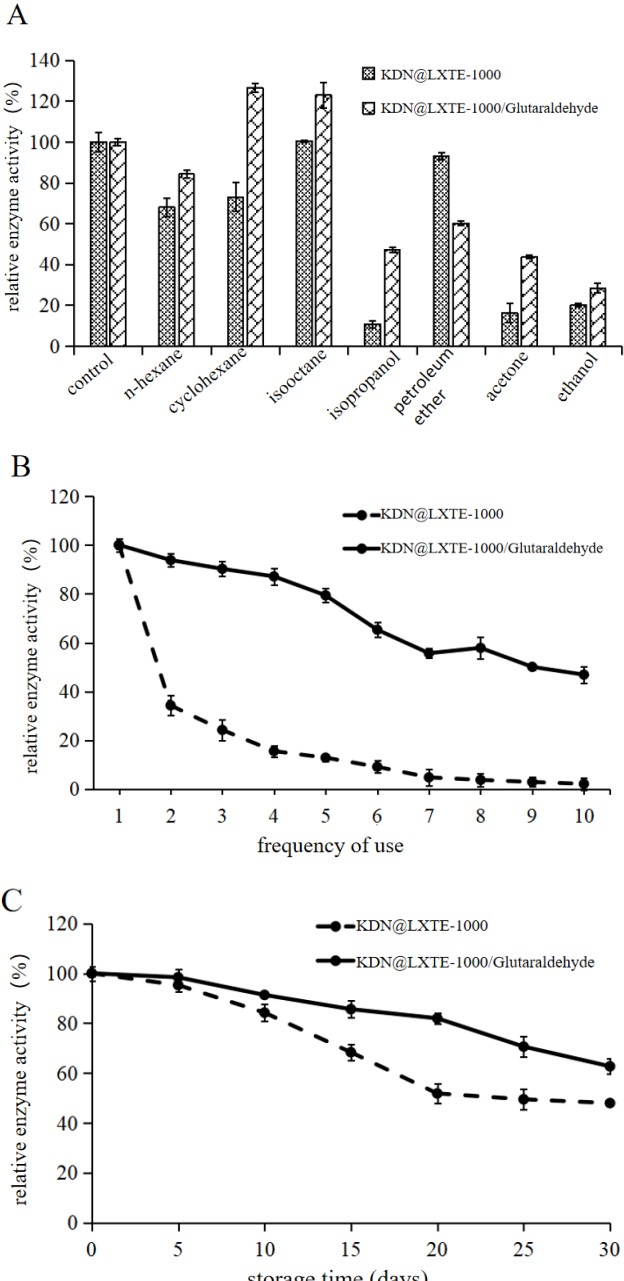

**Figure 5.** Solvent tolerance, reusability, and storage time analysis. (**A**) Solvent tolerance, the 100% of residual activity for KDN@LXTE-1000/Glutaraldehyde, and KDN@LXTE-1000 indicated as the initial lipase activity without solvent addition. (**B**) Reusability. (**C**) Storage time.

Figure 5B shows the reusability of KDN@LXTE-1000/Glutaraldehyde and KDN@LXTE-1000 on the residual activity of the p-NPA hydrolysis. After ten cycles of continuous use, the enzyme activity of KDN@LXTE-1000/Glutaraldehyde preserved more than 46.87% of its initial enzyme activity. After up to eight cycles, KDN@LXTE-1000/Glutaraldehyde is able to retain more than 50% of its initial residual activity. In contrast, the activity of the

immobilized KDN lipase by the adsorption method was only 34.30% of its initial activity after one cycle. The decrease in activity recovery of KDN@LXTE-1000/Glutaraldehyde after ten cycles was most likely due to the washing during separation steps that induced enzyme leaching or desorption of non-covalently bound enzyme molecules from the support. The enzyme reuse rate is an important index to evaluate the immobilization method. The higher the enzyme reuse rate, the more the production cost is reduced.

It can be seen from Figure 5C that after 30 days of storage at 4 °C, the enzyme activity of the immobilized KDN lipase obtained by the adsorption/cross-linking method still retains 62.68% of its initial enzyme activity. Compared with the immobilized KDN lipase by the adsorption method, it only retains 47.93%. Therefore, the immobilized enzyme can resist conformational changes and minimize distortion at the active sites of the enzyme imposed by the aqueous medium [27].

### 2.4. Determination of Myristic Acid and Isopropyl Myristate by GC

The substrate myristic acid and the product isopropyl myristate were detected by gas chromatography. The results are shown in Figure 6. The retention time of myristic acid $R_1 = 5.219$ min, and that of isopropyl myristate $R_2 = 5.893$ min. After calculation, the resolution $R = 4.09 > 1.5$, which meets the requirements.

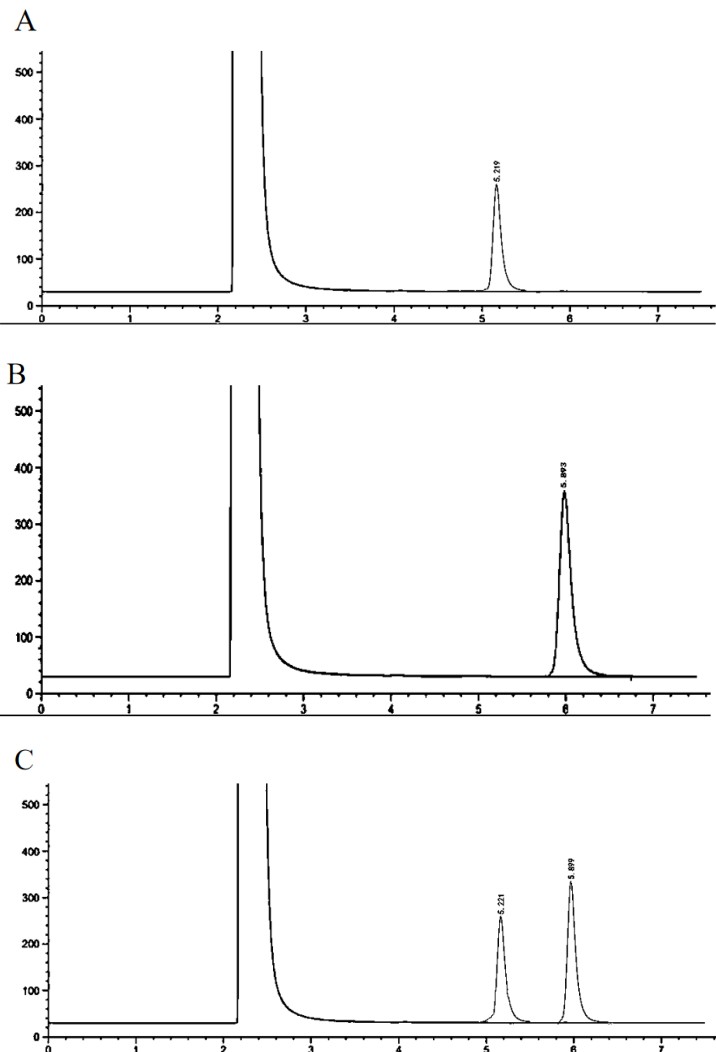

**Figure 6.** GC chart of substrates and products. (**A**) GC chart of myristic acid. (**B**) GC chart of isopropyl myristate. (**C**) GC chart of myristic acid and isopropyl myristate.

### 2.5. Establishment of Solvent System

The influence of the solvent system on the synthesis of isopropyl myristate by adsorption/cross-linking immobilized KDN lipase is shown in Figure 7. The results in Figure 7 show that the yield of isopropyl myristate catalyzed by the six organic solvent systems is not high, and the molar yield of isopropyl myristate in n-hexane is the highest, at 42.99%, the molar yield of cyclohexane is low, at 12.79%. However, the yield of isopropyl myristate in a solvent-free system was higher than that in organic solvents because of the adsorption/cross-linking immobilized KDN lipase. The experiments showed that adsorption/cross-linking immobilized KDN lipase was more suitable for the synthesis of isopropyl myristate in a solvent-free system.

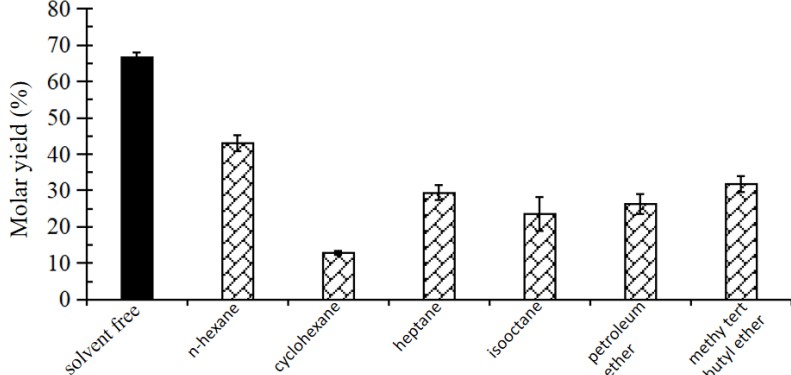

**Figure 7.** Effect of solvent system on the yield of isopropyl myristate synthesized by adsorption/cross-linking immobilized KDN Lipase.

### 2.6. Optimization of Synthesis of Isopropyl Myristate Using RSM

Table 2 shows that $p < 0.0001$, indicating that the model is significant. The model also showed a statistically non-significant lack of fit ($p$-value 0.1539), indicating that the responses are adequate to be employed in this model and that the model satisfactorily fitted the experimental data. Therefore, the model can better explain the influence of four factors (molar ratio of acid to alcohol, reaction temperature, amount of immobilized enzyme, and reaction time) on the molar yield of isopropyl myristate. The quadratic polynomial regression equation is as follows:

$$Y = 51.13 - 3.73A - 1.80B + 10.00C + 4.19D - 1.29AB - 2.69AC - 3.43AD + 1.65BC + 1.13BD - 3.39CD - 5.83A^2 - 8.49B^2 - 4.38C^2 - 2.11D^2$$

It can be seen from Table 2 that the order of influence of the four factors on the molar yield of isopropyl myristate is as follows: amount of immobilized enzyme (C) > reaction time (D) > molar ratio of acid to alcohol (A) > reaction temperature (B). The additional amount of the immobilized enzyme has a very significant effect on the molar yield of isopropyl myristate ($p < 0.0001$), and the reaction time and molar ratio of acid to alcohol have a significant effect on the molar yield of isopropyl myristate ($p < 0.05$). The Design Expert V8.0.6 software was used to analyze the interaction between various factors. The analysis results are shown in Figure 8. In combination with Table 2, it can be seen that the interaction between molar ratio of acid to alcohol and reaction time, the interaction between amount of immobilized enzyme and reaction time have a greater impact on the molar yield of isopropyl myristate.

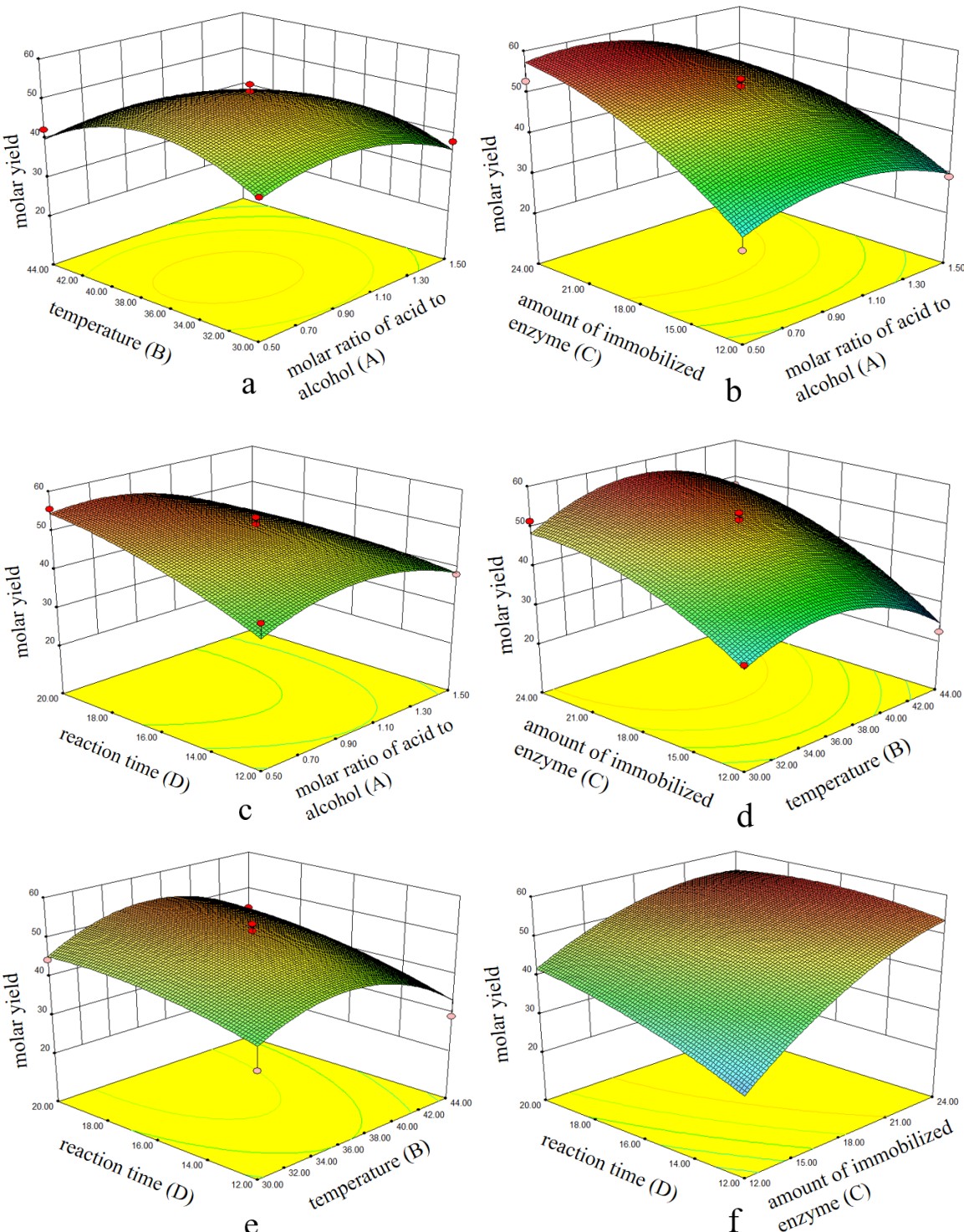

**Figure 8.** Response surface plot and contour plot for synthesis of isopropyl myristate by immobilized enzyme showing significant model terms: (**a**) temperature vs. molar ratio of acid to alcohol, (**b**) amount of immobilized enzyme vs. molar ratio of acid to alcohol, (**c**) reaction time vs. molar ratio of acid to alcohol, (**d**) amount of immobilized enzyme vs. temperature, (**e**) reaction time vs. temperature, and (**f**) reaction time vs. amount of immobilized enzyme.

**Table 2.** ANOVA and fit statistics of the quadratic model for synthesis of isopropyl myristate.

| Model | Sum of Squares | Mean Square | *F*-Value | *p*-Value | Very Significant |
|---|---|---|---|---|---|
| | 2380.09 | 170.01 | 11.66 | <0.0001 | |
| A | 167.03 | 167.03 | 11.46 | 0.0044 | significant |
| B | 38.84 | 38.84 | 2.66 | 0.1249 | |
| C | 1199.40 | 1199.40 | 82.28 | <0.0001 | very significant |
| D | 210.42 | 210.42 | 14.44 | 0.0020 | significant |
| AB | 6.68 | 6.68 | 0.46 | 0.5094 | |
| AC | 28.89 | 28.89 | 1.98 | 0.1810 | |
| AD | 46.99 | 46.99 | 3.22 | 0.0942 | |
| BC | 10.89 | 10.89 | 0.75 | 0.4020 | |
| BD | 5.15 | 5.15 | 0.35 | 0.5616 | |
| CD | 45.83 | 45.83 | 3.14 | 0.0979 | |
| $A^2$ | 220.43 | 220.43 | 15.12 | 0.0016 | |
| $B^2$ | 468.04 | 468.04 | 32.11 | <0.0001 | |
| $C^2$ | 124.55 | 124.55 | 8.54 | 0.0111 | |
| $D^2$ | 29.00 | 29.00 | 1.99 | 0.1802 | |
| Residual | 204.08 | 14.58 | | | |
| Lack of Fit | 179.75 | 17.97 | 2.96 | 0.1539 | non-significant |
| Pure Error | 24.33 | 6.08 | | | |
| Cor Total | 2584.17 | 28 | | | |

The extreme point was obtained by derivation of the regression equation, and the theoretical optimal conditions for synthesis of isopropyl myristate by immobilized enzyme were obtained as follows: The amount of the immobilized enzyme was 23.18% (based on the mass of myristic acid, $w/w$), the reaction temperature was 36.12 °C, the molar ratio of acid to alcohol was 0.67, and the reaction time was 18 h. Therefore, 53 mg of the immobilized enzyme was added, the reaction temperature was 36.1 °C, the amount of myristic acid was 228.4 mg, the amount of isopropyl alcohol was 114 μL, and the reaction time was 18 h. The final molar yield of isopropyl myristate was 66.62%. As shown in Figure S3, during the six times of use, the enzyme activity was not significantly reduced. After six cycles, 79.10% of the relative enzyme activity was retained.

Conventional methods for isopropyl myristate involve the use of a chemical catalyst at high temperatures. However, this results in undesirable changes in the final product with respect to the color, odor, and stability [14]. In the case of isopropyl myristate synthesis, several authors have reported methods for immobilized enzyme (Table 3) [14,24]. The materials used in this paper are inexpensive, easy to operate, stable, and reliable, and we analyzed the purity of the product using gas chromatography.

**Table 3.** Isopropyl myristate synthesis.

| Enzyme | Materials | Conversion Rate | Reusability |
|---|---|---|---|
| Lipase from Bacillus cereus MTCC 8372 | A poly (MAc-co-DMA-cl-MBAm) hydrogel | 66% | 38% (3rd cycle) |
| Novozym 435 | A packed bed reactor | 98.5% (Calculated based on acid consumption) | 50 days |

## 3. Materials and Methods

### 3.1. Enzyme and Chemicals

Commercial lipase was purchased from KDN Biotechnology Co., Ltd. (Qingdao, China). Macroporous resin LXTE-1000 was purchased from Xi'an Lanxiao Technology New Materials Co., Ltd. (Xi'an, China). *p*-Nitrophenol (*p*NP) and *p*-nitrophenolacetate (*p*NPA) were purchased from Sigma-Aldrich, St. Louis, MO, USA. Meanwhile, glutaraldehyde,

polyethylene glycol diglycidyl ether, ethylene glycol diglycidyl ether, and other chemical reagents were purchased from Macklin chemicals (Shanghai, China).

### 3.2. Two-Step Immobilization of KDN Lipase with LXTE-1000 and Glutaraldehyde

Before immobilization, the LXTE-1000 was activated through the following process: LXTE-1000 was first immersed in 95% ethanol for 24 h, stirred continuously, and then washed with pure water until there was no obvious ethanol smell. Pretreatment of lipase was as follows: 6.0 mL of phosphate buffer (50 mM, pH = 7.0) was added to 36.0 mg of lipase, which was then centrifuged at 3500 rpm for 15 min, and 5.0 mL of supernatant was taken for use. Then, added 500 mg of LXTE-1000 to lipase solution (5 mL) and shaked at 37 °C (700 rpm) for 4 h. After that, the suspensions were filtered and washed with the phosphate buffer (50 mM). The immobilized KDN lipase samples were dried at room temperature overnight to remove most of the water, then placed in the incubator at 37 °C for 4 h to remove the remaining water, and labeled as KDN@LXTE-1000.

After adsorption, we added glutaraldehyde, ethylene glycol diglycidyl ether, and poly (ethylene glycol) diglycidyl ether, respectively. Six concentration gradients of 0.1%, 0.3%, 0.5%, 0.7%, 0.9%, and 1.0% (*v/v*) were set for each cross-linking agent. The cross-linking was carried out on a temperature-controlled shaking table at 37 °C, and it was shook at 700 rpm for 4 h. After the reaction was completed, it was filtered and washed with 50 mM PBS buffer solution with pH of 6.8 to obtain the immobilized enzyme, dried overnight at room temperature to remove most of the water, and placed in the incubator at 37 °C for 4 h to remove the remaining water, detect the enzyme activities of different immobilized enzymes, screen the appropriate cross-linking agent, and determine the concentration of the cross-linking agent.

### 3.3. Determination of Lipase Activity

The enzyme activity was determined following a modified method of Liu ZQ [28], using *p*-NPA as a substrate and spectrophotometric detection (WFH-201BJ, Shanghai Jingke Industrial Co., Ltd., Shanghai, China) of the released p-nitrophenol (*p*-NP) at 405 nm. Definition of enzyme activity unit is as follows: under the conditions of 45 °C and pH 8.0, the amount of enzyme required for catalytic production of 1 μmoL pNP within 1 min is defined as 1U.

The hydrolytic activity was calculated according to the following equation:

$$\text{Hydrolytic activity} = \frac{[62.342 \times (X_2 - X_1) - 0.1479] \times V \times N}{m \times t}$$

where 62.342 is the reciprocal of the slope of the standard curve of p-nitrophenol; $X_1$ is the absorbance value of the blank control group; $X_2$ is the absorbance value of the sample group to be tested; $V$ is the reaction volume; $N$ is the dilution ratio of enzyme solution; $m$ is the mass of immobilized enzyme added in the determination of enzyme activity; and $t$ is the reaction time.

The recovery rate of immobilized lipase is calculated as follows:

$$A = \frac{W_0}{W_1 - W_2}$$

$A$ is recovery rate of immobilized lipase; $W_0$ is the activity of immobilized enzyme; $W_1$ is the total activity of added free enzyme; and $W_2$ is the activity of supernatant after centrifugation.

### 3.4. Design for Response Surface Methodology (RSM)

Design Expert V8.0.6 was used to develop experimental design through Box–Behnken design (BBD) to optimize the immobilization conditions. In the experiment of combination of macroporous resin adsorption and cross-linking, the experimental design consists of

3 parameters, which are as follows: cross-linker concentrations (0.1%, 0.3%, and 0.5%), times (2 h, 4 h, and 6 h), and temperatures (25 °C, 30 °C, and 35 °C). In the experiment of isopropyl myristate synthesis, the experimental design consists of 4 parameters which are as follows: the molar ratio of alcohol to acid (0.5, 1, and 1.5), amount of enzyme (12%, 18%, and 24%), reaction temperature (30.0 °C, 37.0 °C, and 44.0 °C), and the reaction time (12 h, 16 h, and 20 h). The regression analysis of the experimental data and response surfaces was performed using analysis of variance (ANOVA).

### 3.5. Biochemical Indexes of Free KDN Lipase and Immobilized Lipase

Biochemical characterization of the free KDN lipase and immobilized lipases included heat stability, pH stability, solvent tolerance, reusability, and storage stability. The optimum conditions for their activity were also determined. The immobilized enzyme and free enzyme were kept at 20.0, 30.0, 40.0, 50.0, 60.0, 70.0, and 80.0 °C for 2 h, respectively, and then the enzyme activities were measured to determine the optimal reaction temperature. The immobilized enzyme and free enzyme were kept at the optimum reaction temperature for 1, 2, 3, 4, 5, 6, and 7 h, respectively, and then their enzyme activities were measured to compare their thermal stability. Incubate the immobilized enzyme and free enzyme in buffer solution with pH values of 4.0, 5.0, 6.0, 7.0, 8.0, 9.0, and 10.0, respectively, for 2 h. The immobilized enzyme was incubated in organic solvents n-hexane, cyclohexane, isooctane, isopropanol, petroleum ether, acetone, and ethanol for 2 h, respectively, and the control group was incubated in 50 mmol/L pH 8.0 Tris-HCl buffer solution for 2 h. The enzyme activity of the immobilized enzymes in the above groups was determined to investigate the organic solvent tolerance of the immobilized enzyme. The immobilized enzyme was assayed for its reusability and recyclability in the reaction ten times under optimum reaction conditions. The optimized conditions were selected to prepare the immobilized enzyme, and the immobilized enzyme was sealed and stored in a 4 °C refrigerator. The activity of the immobilized enzyme was measured every 0, 5, 10, 15, 25, and 30 days, respectively.

### 3.6. Analytical Method of Isopropyl Myristate

The product yield was calculated by gas chromatography (7890A, Agilent Technologies, Inc., Santa Clara, CA, USA) according to the peak area. Chromatographic column: DB-1701 capillary chromatographic column (330 m × 250 μm, 0.25 μm); column box temperature: 200 °C; injection port temperature: 230 °C; detector temperature: 250 °C; split ratio: 10:1; air flow rate: 300 mL/min; hydrogen flow rate: 30 mL/min; tail gas flow rate: 30 mL/min; injection volume: 1 μL; and running time: 7.5 min.

$$Molar\ yield\ of\ isopropyl\ myristate = \frac{Molar\ number\ of\ isopropyl\ myristate\ generated}{Initial\ molar\ number\ of\ myristi\ acid} \times 100\%$$

### 3.7. Statistical Analysis

All the experiments were independently conducted at least three times, and the results were expressed as means ± standard deviations ($n = 3$).

### 4. Conclusions

There are different kinds of lipases from different microorganisms with different catalytic performances. The purified alkaline thermostable bacterial lipase from *Bacillus cereus* MTCC 8372 was immobilized on a poly (MAc-co-DMA-cl-MBAm) hydrogel, and this immobilized lipase is used to achieve esterification of myristic acid and isopropanol in *n*-heptane under continuous oscillations at 65 °C [16]. Some may be immobilized on $Fe_3O_4$/cellulose nanocomposites and showed a high stability of the transesterification/esterification reaction and converted the wasted cooking oil into biodiesel, such as the lipase from *Kocuria flava* [18]. In an engineered *Aspergillus niger,* the *Candida antarctica* lipase B gene is fused with the *Saccharomyces cerevisiae* glycosylphosphatidylinositol protein SED1 and displayed on the surface of *Aspergillus niger*. The highest yields of isopropyl laurate, isopropyl myristate,

and isopropyl palmitate are 79.21, 81.62, and 81.41%, respectively [19]. The immobilized Novozyme 435 was also applied to the enzymatic synthesis of an isopropyl myristate using a packed bed reactor, with a stably operated for at least fifty days and achieved product space–time yield of 26 mM/g/h [16].

Immobilization is essential for the lipase catalyzed reactions. For example, lipase A from *Candida antarctica*, which was immobilized on the natural biopolymer poly(3-hydroxybutyrate-co-hydroxyvalerate) (PHBV) in an aqueous solution, has the ability to attack the sn-2 position of triglycerides [29]. For lipase B, if there are hydrophobic matrix droplets, such as a drop of oil, the lipase will strongly adsorb it and be able to attack the insoluble matrix [30]. These immobilizations are very useful and applicable.

In this study, the macroporous resin LXTE-1000 was selected to immobilize KDN lipase by the adsorption/cross-linking method. This newly developed resin and lipase are combined for enzymatic application for the first time. The optimal immobilization conditions were defined by a single factor experiment and response surface methodology (RSM). The subsequent analysis of variance (ANOVA) showed that the immobilization of KDN lipase was very suitable to be described by a linear model. The optimization conditions for activity recovery and immobilization efficiency are as follows: enzyme 60 mg/g, adsorption temperature 33.9 °C, buffer pH 6.8, and adsorption time 80 min. The concentration of the cross-linking agent glutaraldehyde was 0.46% (*v*/*v*), the cross-linking temperature was 25.0 °C, and the cross-linking time was 157 min. The enzyme activity of the immobilized KDN lipase by the adsorption/cross-linking was 291.36 U/g, and the recovery of the enzyme activity was 9.90%. The stability of the immobilized KDN lipase by adsorption and the immobilized KDN lipase by adsorption/cross-linking were investigated. The investigation found that the thermostability, acid-base stability, organic solvent tolerance, use stability, and storage stability of the adsorption/cross-linking method were better than those of the adsorption method. The optimal conditions for the synthesis of isopropyl myristate catalyzed by the immobilized KDN lipase obtained from *Rhizopus oryzae* in a solvent-free system were determined by a single factor experiment and response surface methodology (RSM): immobilized enzyme 53 mg, reaction temperature 36.1 °C, myristic acid 228.4 mg, isopropanol 114 μL, and reaction time 18 h. The yield of isopropyl myristate was 66.62%. This study explored the potential applicability of immobilized KDN lipase in a solvent-free reaction, which is of great significance for the synthesis of isopropyl myristate. Isopropyl myristate is an important intermediate in the pharmaceutical industry and the cosmetics industry. The combination of the solvent-free reaction system and the immobilized enzyme method is an environmentally friendly and green method. In view of the important role of isopropyl myristate, the method of immobilizing lipase should be further optimized to improve the yield of isopropyl myristate.

**Supplementary Materials:** The following supporting information can be downloaded at: https://www.mdpi.com/article/10.3390/catal13040772/s1. Table S1: Box–Behnken design and its responses for optimization of immobilization conditions of KDN lipase by adsorption method; Table S2: ANOVA and fit statistics of the quadratic model for optimization of immobilization conditions of KDN lipase by adsorption method; Figure S1: Response surface plot and contour plot for adsorption immobilization showing significant model terms; Figure S2: Reusability analysis of immobilization of lipase by adsorption; Figure S3: Reusability analysis of immobilized KDN lipase catalyzed synthesis of isopropyl myristate synthesis.

**Author Contributions:** M.S. (Data Curation, Investigation); Y.X. (Investigation, Methodology); S.C. (Data Curation), W.X. (Weizhuo Xu) (Resources, Supervision and Writing) and W.X. (Wei Xu) (Resources, Supervision, Writing—Review and Editing). All authors have read and agreed to the published version of the manuscript.

**Funding:** This research received no external funding.

**Data Availability Statement:** Data are available upon reasonable request.

**Conflicts of Interest:** The authors declare no conflict of interest.

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
