# Peer review of "Immobilized KDN Lipase on Macroporous Resin for Isopropyl Myristate Synthesis"

_catalysts, doi:10.3390/catal13040772_

Round 1

Reviewer 1 Report

The manuscript is very well written and shown very interesting results and can be accepted after minor revision.

1- A very recent paper on Lipase enzyme application should be cited such as 10.3390/catal12090977.

2- The FTIR analysis of pure macroporous resin and Immobilized KDN lipase on macroporous resin should added to the revised manuscript. 

3- Page 13, line 323, the authors should remove a repeating "The" from the beginning of the sentence.

4- The authors should add a new Table to compare between the activity of the immobilized KDN lipase on macroporous resin for isopropyl myristate synthesis and the other published work.

Author Response

     Thank you very much for your email and reviewer's comments on our manuscript entitled " Immobilized KDN lipase on macroporous resin for isopropyl myristate synthesis ". Your comments are very helpful to revise and improve our manuscript. After carefully studied the reviewers' comments, we have made corresponding corrections with the track mark in the manuscript. Also, we‘ve listed our answers to the reviewer's comment as below in blue italic font. Hope these efforts will make the manuscript acceptable for publication.

     The main corrections in the paper and the responses to editor’s comments are as follows:

Comments from reviewer 1:

  1. A very recent paper on Lipase enzyme application should be cited such as

10.3390/catal12090977 (https://doi.org/10.3390/catal12090977) 

Response to comment #1:

Yes, we’ve cited this paper as Ref #18 in the revised manuscripts.  

  1. The FTIR analysis of pure macroporous resin and Immobilized KDN lipase on microporous resin should added to the revised manuscript.

Response to comment #2:

We are very sorry for this, the experiments had been accomplished for a while and the related experiments materials had not been kept.

  1. Page 13, line 323, the authors should remove a repeating "The" from the beginning of the sentence.

Response to comment #3:

 This repeating “The” had been removed.

  1. The authors should add a new Table to compare between the activity of the immobilized KDN lipase on macroporous resin for isopropyl myristate synthesis and the other published work.

Response to comment #4:

  Table 3, from Line 321, had been added.

Reviewer 2 Report

The first part regarding enzymes and the second about isopropyl myristate do not match each other. Please, try to combine them in some way.

Lines 56-59 – please cite the appropriate reference. Use the superscripts, and add the unit for the melting point.

Define the novelty of your work in the aim of the work. In several papers, the authors used to synthesize this ester with the enzymatic process, mainly lipase-catalyzed reactions.

16 out of 17 references were used in the introduction part. The manuscript has no discussion, hence it can’t be accepted in its present form.

Author Response

     Thank you very much for your email and reviewer's comments on our manuscript entitled " Immobilized KDN lipase on macroporous resin for isopropyl myristate synthesis ". Your comments are very helpful to revise and improve our manuscript. After carefully studied the reviewers' comments, we have made corresponding corrections with the track mark in the manuscript. Also, we‘ve listed our answers to the reviewer's comment as below in blue italic font. Hope these efforts will make the manuscript acceptable for publication.

     The main corrections in the paper and the responses to editor’s comments are as follows:

Comments from reviewer 2:

  1. The first part regarding enzymes and the second about isopropyl myristate do not match each other. Please, try to combine them in some way.

Response to comment #1:

Thanks for your suggestion, we’ve rephrased the lines 56-57 and 76-81 to combine these two parts.

  1. Lines 56-59 – please cite the appropriate reference. Use the superscripts, and add the unit for the melting point.

Response to comment #2:

We’ve added the reference #14 and supplemented the melting point at 3°C.

  1. Define the novelty of your work in the aim of the work. In several papers, the authors used to synthesize this ester with the enzymatic process, mainly lipase-catalyzed reactions.

Response to comment #3:

 Yes, there are several literatures had been used lipase to catalyze this reaction, but the lipases are derived from Bacillus cereus, Aspergillus niger and Novozyme 435. In this work, the KDN lipase was derived from Rhizopus oryzae. So there is significant difference for our study. Meanwhile, the resin used in this research is selected from LXTE-1000, which is also the latest macroporous resin developed with better conjugation performance with enzymes.

  1. 16 out of 17 references were used in the introduction part. The manuscript has no discussion, hence it can’t be accepted in its present form

Response to comment #4:

  Yes, we’ve supplemented the discussion parts in the Conclusion section, and supplemented the references as well.

Round 2

Reviewer 2 Report

The authors addressed all my comments and suggestions. The manuscript has been carefully revised.